# Proof of Concept of a 6-Month Person-Oriented Exercise Intervention ‘MultiPill-Exercise’ among Patients at Risk of or with Multiple Chronic Diseases: Results of a One-Group Pilot Trial

**DOI:** 10.3390/ijerph19159469

**Published:** 2022-08-02

**Authors:** Simone Schweda, Barbara Munz, Christof Burgstahler, Andreas Michael Niess, Inka Roesel, Gorden Sudeck, Inga Krauss

**Affiliations:** 1Department of Sports Medicine, University Hospital, Medical Clinic, 72074 Tuebingen, Germany; barbara.munz@med.uni-tuebingen.de (B.M.); christof.burgstahler@med.uni-tuebingen.de (C.B.); andreas.niess@med.uni-tuebingen.de (A.M.N.); inka.roesel@med.uni-tuebingen.de (I.R.); inga.krauss@med.uni-tuebingen.de (I.K.); 2Interfaculty Research Institute for Sports and Physical Activity, 72074 Tuebingen, Germany; gorden.sudeck@uni-tuebingen.de; 3Institute for Clinical Epidemiology and Applied Biostatistics, University of Tuebingen, 72074 Tuebingen, Germany; 4Institute of Sports Science, Eberhard Karls University, 72076 Tuebingen, Germany

**Keywords:** multimorbidity, comorbidity, health behavior change techniques, chronic non-communicable diseases, cardiovascular diseases, diabetes mellitus, osteoarthritis, overweight, physical activity, exercise

## Abstract

Physical exercise has been shown to be effective in the treatment of non-communicable chronic diseases. However, patients with multiple chronic diseases (multimorbidity) have received little attention in health policy. This pilot trial served as a proof of concept of a 6-months person-oriented exercise intervention for people at risk of or with diagnosed cardiovascular diseases, diabetes mellitus type 2, overweight and/or hip/knee osteoarthritis, regarding effects on health outcomes as well as adherence and safety. The intervention (‘MultiPill-Exercise’) was designed to promote physical exercise participation, considering an individual perspective by addressing personal and environmental factors. Outcomes were assessed at baseline (t0) and after three- (t3) and six-months (t6). The primary outcome was self-reported physical exercise participation in minutes/week comparing t3 and t6 vs. t0. Secondary outcomes included cardio-respiratory fitness (maximum oxygen uptake VO_2_peak during incremental cycling ergometry), isometric peak torque of knee extensors and flexors, health-related quality of life (Veterans Rand 12 with its subscales of perceived general health (GH), mental health (MCS), and physical health (PCS)) and blood levels. Adherence to exercise (% of attended sessions during the first 12-weeks of the intervention) and adverse events were monitored as well. Data were analyzed using a non-parametric procedure for longitudinal data, estimating rank means (M_Rank_) and relative treatment effects (RTE) as well as linear-mixed effect models for parametric data. The primary endpoint of physical exercise participation was significantly higher at t3 and t6 compared to baseline (t3 vs. t0: M_Rank_ = 77.1, *p* < 0.001, RTE: 0.66; t6 vs. t0: M_Rank_ = 70.6, *p* < 0.001, RTE = 0.60). Improvements at both follow-up time points compared to t0 were also found for relative VO_2_peak (t3 vs. t0 = 2.6 mL/kg/min, *p* < 0.001; t6 vs. t0 = 2.0 mL/kg/min, *p* = 0.001), strength of knee extensors (t3 vs. t0 = 11.7 Nm, *p* = 0.007; t6 vs. t0= 18.1 Nm, *p* < 0.001) and GH (t3 vs. t0 = 16.2, *p* = 0.003; t6 vs. t0 = 13.4, *p* = 0.008). No changes were found for MCS, PCS and for blood levels. Overall exercise adherence was 77%. No serious adverse events were recorded. Results of this pilot trial represent a first proof of concept for the intervention ‘MultiPill-Exercise’ that will now be implemented and evaluated in a real-world health care setting.

## 1. Introduction

Physical inactivity is one of the major public health problems of the 21st century [1]. Even though evidence for the effectiveness and the various health benefits of physical activity (PA) exists, about one third of the world’s population does not achieve health-oriented PA recommendations [2]. To address this problem the WHO introduced the Global Action Plan on Physical Activity (GAPPA) 2018–2030 with the aim to reduce the prevalence of physical inactivity by 15% by 2030 [3]. Still, the proportion of insufficiently active people in Europe has increased in recent years. Currently, 46% of Europeans never exercise [4]. Physical inactivity is a major risk factor for several non-communicable chronic diseases (NCDs) such as Diabetes mellitus type 2 (DMT2), cardiovascular diseases (CVD), hip and/or knee osteoarthritis (OA) as well as overweight (OW) and obesity (OB), all increasing the risk of premature mortality [1,2,5,6]. Due to overlapping risk factor profiles, patients diagnosed with only one of these conditions are rare–typically, they present multiple chronic conditions simultaneously. The presence of two or more diseases at the same time is referred to as multimorbidity [7,8]. Multimorbidity is a common health problem whose prevalence further rises with age [7,9]. About two thirds of adults over 50 years suffer from at least two chronic conditions [9,10]. Since demographic change augments the proportion of the elderly within the population, multimorbidity will also become more prevalent. A recent study showed that people older than 75 years suffering of multimorbidity lived 81% (median 5.2 years) of their remaining years of life with disability [11]. Apart from decreases in quality of life, polypharmacy, increased health care use and rising health-related costs are major consequences [7,9]. About 70–80% of health care costs in the European Union are spent on chronic diseases [12].

Nevertheless, the treatment of multimorbidity is often uncoordinated and burdensome [13] and conservative treatment options are sparse. Deficits in managing chronic conditions in the current health care system have been outlined [12,13]. To meet this problem, several disease management programs have been developed in recent years [14]. Yet, most of these programs focus on single-disease approaches and therefore do not meet the demands of multiple chronic conditions [8,15]. This also applies to physical exercise interventions as being a relevant component of disease management programs. Moreover, it is not unusual for people with multiple chronic conditions to be excluded from exercise programs, due to the fear of comorbidity-associated adverse events [16,17], although the American College of Sports Medicine (ACSM) explicitly stated that the health benefits of PA almost always outweigh its risks [18]. Despite given evidence for the effectiveness and recommendation of PA in many NCDs such as metabolic and cardiovascular diseases as well as musculo-skeletal disorders [6], PA has not yet been established as part of disease management programs aiming at the prevention and treatment of multimorbidity. This is particularly alarming, because people who suffer from NCDs are even less physically active than healthy adults [19,20,21].

Reasons for physical inactivity include fear of complications (pain, stiffness, fatigue), the perception of inability to exercise as well as lack of knowledge on the benefits of exercising [22,23]. These barriers as well as disease-specific conditions, personal context factors (i.e., motives for exercise participation, social support) and environmental factors (i.e., access to exercise facilities) have to be kept in mind when designing a PA intervention program for patients suffering from multimorbidity [24,25]. In order to achieve a long-term and regular health-promoting behavior, holistic approaches are necessary that do not only include physical training, but also foster the development of physical activity related health competences to enable patients to initiate and maintain health-enhancing PA [26,27]. For optimal health outcomes, even some physical activity is better than none, yet more PA is better [28]. According to the World Health Organization 2020 guidelines on physical activity, people living with chronic conditions should undertake 150–300 min of moderate-intensity, or 75–150 min of vigorous-intensity aerobic PA or an equivalent combination of both intensities. Aerobic training should be supplemented with muscle-strengthening activities for all major muscle groups on two or more days a week at moderate or greater intensity. Additional health benefits are expected when the minimum dose is further increased [28].

To summarize, physical inactivity is a major risk factor for many NCDs and people suffering from multimorbidity are specifically vulnerable as they are even less active than the norm. As health care programs including physical exercise interventions primarily focus on single disease management, a holistic approach specifically designed for people being at risk or suffering from multimorbidity is needed. Therefore, the aim of this study was to conduct and to proof the concept of a person-oriented exercise intervention (‘MultiPill-Exercise’) aiming to enable participants with multimorbidity to engage in regular exercise according to the PA guidelines.

The primary objective was to evaluate self-reported levels of physical exercise participation throughout the intervention. In addition, cardio-respiratory fitness, muscular strength, health-related quality of life (HRQoL) and blood levels were evaluated. The proof of concept was further related to training adherence and safety of the intervention.

## 2. Materials and Methods

### 2.1. Study Design and Setting

This study was conducted as a longitudinal one-group pilot trial. The study was approved by the local ethics committee of the University Hospital of Tuebingen and registered at the German clinical trial register (DRKS00016702). All participants provided written consent to participate in the study. The study was conducted at the University Hospital Tuebingen, Germany in two recruiting periods. The maximum time period between baseline assessment and start of the intervention was two weeks. The overall study period was 24 weeks. Follow-up measurements were conducted after 3 months (t3) and after 6 month (t6). Details are depicted in Figure 1.

### 2.2. Participants

Volunteers were recruited via advertisements in the local newspaper, flyers, institutional newsletters and general practitioners.

We screened for insufficiently active adults aged 18 years or older being at risk or suffering from multiple chronic conditions. Specific inclusion and exclusion criteria for study participation are outlined in Table 1.

### 2.3. Sample Size Calculation

NQuery Advisor software (GraphPad Software DBA Statistical Solutions, San Diego, CA, USA) was applied for estimating the sample size. Based on a one-group, two-sided t-test with a medium effect size according to Cohen’s d = 0.50 [29], α = 5% and a power of 80% a sample size of *n* = 34 participants for final data analysis was calculated. Considering an estimated fail-out rate of 50% between screening and final inclusion, we aimed to include 68 subjects in the medical screening.

### 2.4. Person-Oriented Physical Exercise Intervention ‘MultiPill-Exercise’

‘MultiPill-Exercise’ relates to the biopsychosocial framework of the International Classification of Functioning, Disability and Health (ICF) [30]. It incorporates a biomedical perspective by dosing exercises according to the PA recommendations of the WHO, with the aim to improve body structures and functions. This in turn counteracts existing limitations in activities and participation and thus can ease the burden of the disease. To achieve physical activity-related health effects ‘MultiPill-Exercise’ includes an individualized training based on the individual’s physical capacity. For this purpose, the training prescription for endurance exercises is determined on basis of the cycling ergometry of the exercise pre-participation examination at the participant’s heart rate (HR) and power output (PO). For moderate exercises HR and PO at 110% of LT1 and for vigorous intensity exercises HR and PO at 90% maximum power output (Pmax) (for more details. Table A1) were used. Training intensity was monitored by individual heart rate responses or perceived exertion (BORG CR10 0–10 point [31]). Next to aerobic exercises 2–3 times/week for 30–60 min, machine-based strength training in the gym (1 x/week), functional strength training at home (1 x/week) were included in the intervention. However, this biomedical perspective cannot be effective until it is possible to encourage individuals to engage in regular health-promoting exercise and to support them in continuing to do so independently. The physical activity-related health competence (PAHCO) with its sub-domains—(1) ‘movement competence’; (2) ‘control competence’; and (3) ‘self-regulation’—is an important personal context factor enabling the individual to exercise in a health-enhancing manner [26,27]. ‘MultiPill-Exercise’ comprises specific elements to foster each of the subdomains. Therefore, educational sessions and workshops, individual counselling sessions and disease specific offers were included in the intervention. Motor abilities and skills as well as body awareness as relevant (1) ‘movement competences’ were supported in the exercise sessions. Four ‘movement teasers’ each lasting 60 min were included to make participants familiar with different types of exercises. Knowledge on training prescription and health effects of exercises, feedback on exercises and the integration of the gained knowledge in the home-based exercises were implemented to improve (2) control competence. When it comes to (3) self-regulation, motivational and volitional determinants are of particular importance. The intervention therefore put a specific focus on enabling participants to find individual motivation for exercising and suitable activities and to use techniques to ease behavior change. Weekly training schedules and logs served as further support (Figure 2). A more detailed description of the intervention concept has been described elsewhere [32,33].

All components, their dosages and their theoretical foundation are outlined in Table A1.

The intervention delivery was divided into a 12-week supervised phase with regular sessions at the University Hospital and a subsequent 12-week self-directed phase. In the first phase endurance training increased steadily with the goal of eventually meeting national PA recommendations at its end. From week 13 on, participants were encouraged to maintain the same amount of physical exercises as before; however they were to self-organize their exercises in local fitness or recreational centers, non-institutional organizations or in their own environment.

Due to the COVID-19 pandemic, the intervention in this study had to be adapted for the second recruitment group from week nine onwards. Group trainings and educational sessions as well as strength training instructions were provided in a digital format.

### 2.5. Data Collection and Outcome Measures

Participant’s characteristics, physical pre-participation examinations and performance tests were assessed on-site. Patient-reported outcomes were assessed using an online questionnaire (Questback GmbH, Cologne, Germany, that was sent immediately after the on-site assessments. If the questionnaire was not answered within one week, two reminders were sent, each three days apart. Individual questionnaires were available for a maximum of two weeks. The measurements were taken at baseline (t0), after three- (t3) and after six-months (t6). Due to COVID-19-related contact restrictions, no on-site measurements were taken in group 2 at t3.

#### 2.5.1. Baseline Characteristics and Exercise Pre-Participation Screening

The baseline screening included a medical anamnesis to assess eligibility criteria as well as comorbidities, family history of disease, current medications, lifestyle habits and (other) medical conditions. Fasting blood samples were collected to check for abnormal values. Questionnaires to assess the risk or disease status of the diseases of interest [34,35,36] were completed by the participants. Exercise pre-participation screening included the determination of resting heart rate, BP and ankle-brachial index (ABI; determines BP on all limbs), measured after a five-minute rest using the BOSO-ABI-System 100 automatic blood pressure monitor (Bosch + Sohn GmbH & Co. KG, Jungingen, Germany), a resting ECG (custo cardio 300, custo software, custo med GmbH, Ottobrunn, Germany) followed by an ECG-guided incremental cycle ergometer test (more details in Section 2.5.3). Reasons for immediately stopping ergometry included chest pain, systolic BP > 200 mmHg, intolerable dyspnea and cramps.

#### 2.5.2. Primary Outcome Measure: Physical Exercise Participation

*Physical exercise participation* was assessed via the Physical activity, exercise and sport questionnaire [Bewegungs- und Sportaktivitaets-Fragebogen (BSA-F)]. The self-reported retrospective (four weeks) instrument has high construct and criterion validity [37]. It is based on the Frequency-Intensity-Time-Type (FITT) approach [38]. The questionnaire also distinguishes between PA in daily life (including work and leisure time as well as cleaning and gardening) and physical exercise (physical activities that are performed for their own sake, such as swimming, walking etc.). Only the overall time spent with physical exercise per week was calculated according to Fuchs et al. [37] and included in the final analysis.

#### 2.5.3. Secondary Outcome Measures

##### Physical Performance Measures

*Cardio-pulmonary outcomes*. Cardiopulmonary exercise testing (CPET) was conducted on a cycle ergometer (Ergoline Ergoselect 200 or Lode Excalibur Sport). The incremental test-protocol started with 25 W for women and 50 W for men and increased 25 W for both women and men each 3 min until exhaustion. CPET was performed using MetaLyzer 3B-R2 and MetaSoft Studio (Cortex Biophysik GmbH, Leipzig, Germany) to determine peak oxygen uptake (VO_2_peak) and oxygen uptake at the ventilatory anaerobic threshold (AT) as well as the lactate threshold (LT). Capillary blood samples were taken from the hyperemic earlobe before the start of the CPET, at the end of each performance level and five minutes after the end of the test. Performance at the individual LT 1 (LT 1: first increase of lactate) was determined for training prescription only. ECG was used for monitoring and to assess heart rates at the end of each stage. Pmax was defined as maximum power in W recorded during the test.

*Muscular strength measures*. Maximum isometric torque and force ratios for knee extensors and knee flexors were quantified in standardized starting positions on DAVID power equipment (Schupp GmbH & Co. KG, Dornstetten, Germany, F 200 Leg Extension & F 300 Leg Curl). Each participant performed a 10-min warm up on a cycle ergometer at 30 W prior to strength measures. One test trial was performed before two readings of maximum force were taken. In the case of deviations > 7 Nm, a third test was carried out. The best test was used for evaluation. For data analysis, mean values of both legs were used.

*Metabolic measures.* Venous blood was collected in the morning after a fasting period of 12 h. Glycated hemoglobin (HbA1c) as well as glucose and cholesterol, triglycerides, HDL and LDL were evaluated.

*Anthropometric measures.* Body weight and height were assessed subsequent to the blood sampling. Body weight was used as an outcome measure and body mass index (BMI), defined according to the WHO, was calculated to characterize participants at baseline.

*Health-related Quality of Life (HRQoL).* Patient-reported HRQoL was recorded using the validated and adapted German version of the Veterans Rand 12 (VR12) [39]. This instrument covers global health status. It is divided into eight health domains: General Health (GH), which can also be used as a one-item scale [40,41], physical functioning, pain, role limitations because of physical problems, mental health, social functioning, role limitations because of emotional problems and vitality. Two standardized summary scores that are related to the general US population of 1990, the physical component scale (PCS) and the mental component scale (MCS) as well as the score for GH as a one-item scale were calculated [40]. Scores range from 0–100. Higher values indicate better functional health and well-being.

*Adherence* was recorded for the first 12-weeks for all supervised training sessions including group trainings and strength machine-based trainings by the instructors of the respective sessions. To assess home based training sessions, weekly standardized training logs were requested from the participants via E-Mail and subsequently analyzed. As the second intervention phase was not supervised and the participants were asked to perform all exercises on their own, adherence was not calculated for the second 12-weeks.

*Adverse events* were recorded by the instructors of the supervised training sessions. Participants were further instructed to comment on any unexpected exercise-related adverse event in a separate section of the training log.

### 2.6. Statistical Methods

#### 2.6.1. Data Analysis

Statistical analyses were conducted with SPSS Statistics version 26 (IBM, USA) and R version 4.0.4. Statistical significance was set as *p* < 0.05. Descriptive statistics are presented in means ± standard deviation (SD) and median ± interquartile range (IQR), absolute (*n*) and relative (%) values. Data were analyzed using linear-mixed effects models (LMM) and non-parametric longitudinal procedures as outlined below. For pairwise comparisons between time points, post-hoc analyses with Bonferroni adjustment were applied in both, non-parametric and parametric models. For the LMMs, effect sizes (ES) were calculated according to Cohen’s d and interpreted as small (0.2 to <0.5), medium (0.5 to <0.7) or large (>0.8) [29].

*Analysis of primary outcome.* Data for physical exercise participation violated the assumption for LMMs of normally distributed residuals and were therefore analyzed with a non-parametric procedure for longitudinal data (R package *nparLD*). This rank-based approach produces estimates of the rank means and relative treatment effects (RTE). The RTE can be interpreted as follows: a randomly chosen observation from the whole dataset results in a smaller value than a randomly chosen observation from the measurement time point with an estimated probability (in %) of RTE × 100. For details, please refer to Noguchi, et al. [42].

*Analysis of secondary outcomes.* Data of all secondary outcomes were analyzed with linear mixed-effects models using the Restricted Maximum Likelihood (REML) approach. We did not account for the nesting of individuals within the incidental level of recruiting groups due to the low sample size at higher levels. Fitting a 3-level null model (random intercepts-only LMM) including subjects and group as random effects resulted in a non-significant group variance component. Introducing the recruitment group as a fixed effect yielded non-significant results. We therefore fit 2-level models with a random intercept for subject, and time (treated as categorial with levels baseline (t0), t3, t6) as a fixed effect. As covariance structure for repeated measures, a first-order autoregressive covariance structure was chosen.

*Analysis of adherence.* To evaluate adherence to exercise, training logs were analyzed. Reported and scheduled exercise sessions for endurance and strength training (machine-based and functional training) were cross-checked, and the percentage of adhered was expressed in percentage of all scheduled training sessions. Strength training was prescribed two times a week for the whole intervention period. The maximum that could be achieved was 100%, even if participants outperformed their scheduled training program. Missing plans were considered with 0% adherence. The number of missing data sets was recorded. Where provided, reasons for missed training sessions were extracted from the training logs and categories were formed to summarize them.

*Analysis of adverse events.* Reported adverse events were categorized into serious adverse events and other adverse events [43].

#### 2.6.2. Handling of Missing Data

*Primary outcome.* Unlike LMMs procedures, the *nparLD* approach used for the analysis of the primary outcome applies list-wise deletion in the case of missing data on any of the model variables, which is known to yield biased estimates unless data are missing-completely-at-random (MCAR). The data seemed unlikely to be MCAR [44]. We therefore assumed a missing-at-random mechanism (MAR) and applied multiple imputation (MI) generating 100 imputed datasets using the R package *Amelia* [45]. Missingness proportions of physical exercise data were 8% at t3 and 21% at t6. Variables included in the MI model were sex, age, weight, body fat, VO_2_peak, physical exercise participation, as well as auxiliary variables sports history (assessed via three items on exercise memories (exercise memories at 20–30 years of age, remembered sportiness in childhood and adolescents, remembered physical education experiences [46]) and depression score. The fit of the MI model was checked visually using distributional plots and calculating descriptive statistic. A sensitivity analyses for the primary outcome was conducted without imputed data.

## 3. Results

### 3.1. Recruitment

Overall *n* = 224 subjects were screened for eligibility, *n* = 57 participants were invited for medical screening of which *n* = 16 (28%) were excluded with reasons (Figure 3). *N* = 41 (72%) participants were eligible and underwent the exercise pre-participation screening and baseline examinations (no exclusions). Two persons withdrew their study participation consent before the start of the intervention. Finally, *n* = 39 (68%) participants were included in the study of which *n* = 6 were lost to follow-up (15% of included subjects).

### 3.2. Baseline Characteristics of the Participants

Overall, *n* = 39 (females: 27; males: 12) participants (age = 55.2 ± 10.3 years; BMI = 31.1 ± 3.0 kg/m^2^) were included in the study. All participants were German citizens. Baseline characteristics and frequencies of the prevalence (risk for) each of the included diseases are outlined in Table 2. *N* = 12 (31%) participants showed one manifested disease which was accompanied by at least one risk factor. Two simultaneous manifested diseases were reported by *n* = 18 (46%) participants, three diseases by *n* = 7 (18%) and *n* = 1 (3%) participant showed four manifested diseases.

### 3.3. Primary Outcome Measure: Physical Exercise Participation

Analyses revealed that participants showed a significant time effect of self-reported physical exercise participation from baseline to t3 and t6 as displayed in Table 3. Post-hoc analyses indicated that differences were significant from baseline to t3 (Rank Mean (M_Rank_) = 77.1, *p* < 0.001, relative treatment effect (RTE) = 0.66) and baseline to t6 (M_Rank_ = 70.6, *p* < 0.001, RTE = 0.60). A non-significant decrease for physical exercise was found between follow-ups t3 and t6.

Figure 4 displays the median physical exercise participation distribution for all three time points as well as post-hoc differences. Sensitivity analyses without imputed data showed similar results (Appendix A). The median level of reported physical exercise participation at baseline was 0.0 min/week and increased to 215 min/week at t3 and then declined again to 139 min/week at t6. Overall, 24 (61%) participants exercised in line or beyond the PA recommendations at t3. At t6, 15 (38%) participants reported a physical exercise participation in line with the PA recommendations. Seven participants (18%) reported not to perform any physical exercise after 6-month, while the remaining participants stated an increase in physical exercise participation compared to baseline. Individual changes in physical exercise participation are displayed in Figure A1.

### 3.4. Secondary Outcomes 

All secondary outcomes are displayed in Table 4.

#### Physical Performance Measures

*Cardio-pulmonary outcomes*. Relative VO_2_peak showed a statistically significant improvement at both follow-up time points vs. t0 with medium effect sizes. VO_2_peak increased significantly from baseline to t3 (small effect) but not towards t6.

*Muscular strength measures*. Improvements for both, isometric peak torque extension and flexion was found for both follow-ups. However, only the increase for extension was statistically significant, both for t3 and t6, indicating small effects.

*Metabolic Measures.* For none of the metabolic outcomes, statistically significant results were obtained. Blood lipid profiles were unchanged for all time points. No change was found for fasting blood glucose as well and only small but statistically non-significant changes for HbA1c levels were observed.

*Anthropometric measures.* Body weight decreased, yet the results were not statistically significant.

*Health-related Quality of Life (HRQoL)*. No difference of the PCS and MCS scores were observed from baseline to t3 and t6. GH showed a statistically significant increase and medium effect from baseline to t3 and t6.

*Adherence.* Mean adherence to the prescribed training sessions of the first 12-weeks (*n* = 38) was 79 ± 9% for endurance exercise sessions 74 ± 14% for strengthening exercise sessions. Overall, 8% of data sets were missing. Stated reasons for a missed training session were elevated blood pressure (*n* = 2), shortage of time (*n* = 87), acute illness (*n* = 56), lack of motivation (*n* = 5) and perceived overload (*n* = 15). In some cases, several reasons were given at the same time. Because of the onset of the COVID-19 pandemic, adherence was further differentiated for the two recruitment groups separately (Table A2).

*Adverse events*. Over the entire period of 24 weeks, no serious adverse event was recorded. One participant suffered from dizziness and sickness after training once. Other training-related adverse events throughout the intervention were delayed onset muscle soreness (*n* = 42), general pain (*n* = 18), leg pain (*n* = 5), hip pain (*n* = 15), knee pain (*n* = 29), foot pain (*n* = 7), lower or upper back pain (*n* = 16) as well as shoulder pain (*n* = 7). In addition, muscle cramps were reported once. None of the reported adverse events put the participant in danger or required medical or surgical intervention.

## 4. Discussion

This pilot study served as a proof-of-concept for a lifestyle intervention program to promote physical exercise participation in people at risk or with manifested multiple chronic diseases. The results of the study revealed statistically significant effects after the 24-week ‘MultiPill-Exercise’ intervention relating to physical exercise participation, cardiorespiratory fitness (relative VO_2_peak, VO_2_peak), muscular strength of the lower body as well as the perception of general health among people at risk or with manifested multiple NCDs. The adherence towards the intervention was good and no severe adverse events occurred.

Results of this study showed a significant increase of self-reported physical exercise participation. Considering the WHO-GAPPA target of a 15% relative reduction in the prevalence of inadequate physical activity [3], this trial can be considered successful with 61% of participants at t3 and 38% of participants at t6 exercising according to or beyond physical activity recommendations. The decrease in physical exercise participation after the supervised period might have had several reasons: (1) The first group started with the second 12-week period in November. The months from October to March have been found to increase time spent sedentary on the northern hemisphere [50]. Poor weather conditions, as common in these months, have been stated as barriers to PA [22,51]; (2) The second intervention group had to be switched to online training already in the 9th week of the first intervention phase due to COVID-19-related contact restrictions. In addition, local sports clubs and also public sports facilities were not accessible for this group in the second phase.

The increase in physical exercise participation is consistent with other studies. An RCT by Lo, et al. [52] demonstrated that a 12-week individualized aerobic exercise training in a rehabilitation center combined with motivational interviewing among middle-aged people with multimorbidity increased total PA by 49%. Weinstein, et al. [53] reported a significant increase in self-reported PA (Human Activity Profile) after 10 weeks of supervised treadmill walking in patients with pulmonary arterial hypertension. An increase in physical exercise participation in line with our study results were also stated for a psychological group intervention on physical exercise and health (MoVo-Concept). The randomized controlled trial (RCT) was conducted with 220 initially inactive participants with orthopedic conditions (arthritis, chronic back pain etc.) in an orthopedic rehabilitation clinic [54]. Participants of the intervention group performed 156 min/week of physical exercise six weeks after discharge from the clinic. After 6-months, physical exercise participation dropped to 91.7 min/week and remained stable at 96.1 min/week after 12-months [54]. Unlike the cited studies, our study was not conducted in the context of rehabilitation. Thus, our participants were exposed to the demands of everyday life from the onset, which might have caused increased difficulties in attempting to incorporate exercises into daily routines compared to a rehabilitative setting. On the other hand, the setting of the intervention can be seen as strength of the intervention, as the participants had no further barrier regarding integration of physical exercise into everyday life after the end of the intervention. In addition, in our study, only moderate to vigorous aerobic recreational physical exercises (walking, swimming, cycling, etc.) were considered for calculation of times spent exercising. As the primary aim of the intervention ‘MultiPill-Exercise’ was to enhance physical exercise participation rather than an overall active lifestyle, transportation-related activities, which, in most studies, are included in the calculation of moderate to vigorous PA [55,56], were not taken into account. This should be considered when interpreting the results.

Cardio-respiratory fitness (CRF) has been discussed as an important outcome [57,58]. The results of this study demonstrate increases in VO_2_peak of 10–13.5%. These results are clinically relevant. A recent study showed that an increase in CRF of 10% should be aimed for in exercise trials for a risk reduction in all-cause mortality of 10–25%.

The weight reduction in this study revealed a statistically non-significant average decrease of 1.6 kg after 24 weeks. This range of weight loss is in line with other studies [52,59]. A recent meta-analysis revealed that exercise interventions for obese people with a mean duration of 20.3 weeks resulted in average weight losses of −0.05 to −1.01 kg [59]. It has been argued that in the context of weight loss, exercise training alone is not an effective therapy. By contrast, a hypocaloric diet in combination with exercise training seems most effective in the management of obesity [59,60,61]. This also holds true for metabolic outcomes. A decrease in lean fat mass and a reduction of abdominal fat has been associated with favorable changes in insulin sensitivity, fasting blood glucose and HbA1c [62,63,64]. In our study, metabolic parameters showed a slight reduction, however, there were no statistically significant changes. This is in line with other studies. A 12-week RCT including 79 obese adults compared the effects of exercise alone with those of a hypocaloric diet and a combination of the two. Triglycerides, glucose and insulin showed a significant and similar decrease in the diet and combination groups, but not in the exercise-only group. Moreover, a significant increase in HDL was only found in the combination group [65]. Although no metabolic changes occurred in our study, muscular strength, especially of the knee extensors (increase of 10% at t3 and 16% at t6), was significantly improved. This increase is in line with recent research. In participants with knee OA and comorbid obesity, 12-week non-weight bearing and weight bearing quadriceps-strengthening exercises both resulted in similar strength gains of 8% and 15% [66]. Results of an RCT on patients with hip OA who performed a 12-week exercise therapy concept, showed a significant increase in hip muscle strength of 8% and 9% for abduction and adduction and of 10% and 6% for hip extension and flexion [67].

Besides physical health outcomes, the positive effects of PA on HRQoL could be demonstrated, even despite other comorbidities [6,16,62,66,68]. A study on weight maintenance in adults with obesity compared four groups: exercise in accordance with the international PA guidelines, the medication liraglutide, a combination of both, and placebo. Only the groups with PA showed long-term improvement in general health perception and emotional well-being [62]. In our study, self-reported general health strongly increased, particularly between t0 and t3 (increase of 16.4 points). The PCS increased (t3: 1.6 and t6: 1.5 points) over the study period, even though not in a statistically significant manner. The MCS increased by 3.1 points throughout the first intervention phase but, unlike expected, decreased below the initial level (−0.7 points) after t6. Reasons for the decrease might partly be explained with the consequences of the onset of COVID-19 for group 2. Contact restrictions and social distancing are known to have led to increased stress and depression [69,70]. To assess the onset of the COVID-19 pandemic in this population, an additional retrospective online survey was conducted. A more detailed explanation on possible consequences was published by our study group elsewhere [71].

Adherence to the PA recommendations is crucial for health benefits [25]. However, adherence has been shown to be even lower amongst people with NCDs when compared to the general population [19,20,21]. The intervention ‘MulitPill-Exercise’ was designed to ease adherence by reducing exercise barriers such as time limitations. Therefore, at least one exercise session per week was planned to be completed at home. In addition, a special focus was placed on individual preferences and the inclusion of motivational elements, which is considered an additional facilitator for long-term physical exercise participation [52,68]. Furthermore, initial supervision, individualized physical exercise prescriptions and the combination of both aerobic and strength training has been proven to enhance adherence, especially in patients with chronic conditions [6]. These recommendations were also considered in the design of the intervention and resulted in an adherence rate of 74% for strengthening exercises and 79% for endurance exercises during the first intervention phase. In a recent review the average adherence rate for clinic- and home-based PA interventions among cancer, cardiovascular disease or diabetes patients was reported to be 77% [68], indicating that adherence rates observed in our study are within an average range. Adherence was not evaluated for the second phase, as no supervised sessions were included. The training plan only specified durations in line with the minimal exercise recommendations.

This study showed no serious adverse events. The initial medical screening and pre-exercise participation examination, as recommended by European Federation of Sports Medicine Associations (EFSMA) [72] along with the integration of PAHCO including BCTs, showed that exercise is safe in this population and outweigh the risks, as already described elsewhere [18,73,74].

### Limitations

This study is not without limitations. First of all, this study is a one-group trial. An overestimation of the results cannot be excluded due to the missing of a control group. Most importantly, our data do not allow objective quantification of exercise participation, which was most likely lower than reported: A social desirability recall bias and the tendency to overestimate PA time has been discussed when using self-reported questionnaires [75]. To circumvent this problem, objective assessments, such as accelerometers, have been increasingly used to monitor activity levels [50,75]. However, a review showed that PA questionnaires are valid and can be recommended, especially because of the low cost and time requirements [75]. Nevertheless, larger studies should, if possible, use both objective measurements and questionnaires. In this context, training supervision, especially for home-based training sessions, must also be considered critically. The intensities for individually performed sessions were only controlled by the patient’s own measuring devices or a subjective exertion scale (BORG). Thus, the extent to which the actual intensities and functional correctness corresponded to the targeted intensities could not be fully monitored.

Moreover, a potential volunteer bias needs to be considered, as participants were recruited via newspaper and public media, and those who applied for participation did not necessarily represent the population of multimorbid patients. They are the most motivated ones, most probably having a compliance much better than the average. A possibly higher compliance of the participants due to initially higher motivation cannot be excluded. Due to the COVID-19 contact restrictions, no follow-up measurements for the second group were possible and therefore physical data for t3 are missing for this group. Long-term PA participation cannot be addressed as there was no follow-up period after six-months. Further studies should consider longer follow-up periods to better account for long-term effects and, among others, seasonal influences.

## 5. Conclusions

At a political level, calls for conservative treatment methods for people with multimorbidity are increasing. This study contributes to gain further understanding regarding the delivery of exercise interventions for individuals with NCDs. The main findings of the trial can be summarized as follows: (1) Participants of ‘MultiPill-Exercise’ were able to engage in regular physical exercise participation and physical exercise could be maintained for six months; (2) Health benefits, especially with regard to cardiorespiratory fitness, muscular strength and perceived general health were achieved. However, for optimizing the effectiveness of the intervention on body weight and metabolic outcomes, dietary issues need to be considered in more detail; (3) Satisfactory adherence was achieved during the comprehensive supervised intervention period, highlighting the general feasibility of exercise interventions in this specific patient group; (4) No severe adverse events were reported in the context of the intervention, suggesting that physical exercise can be used as a safe treatment option for patients with multimorbidity. Based on the successful proof of concept, a larger-scale randomized controlled trial in the field of health care will review these results and establish transfer routes for successful implementation in regular health care practice.

## Figures and Tables

**Figure 1 ijerph-19-09469-f001:**
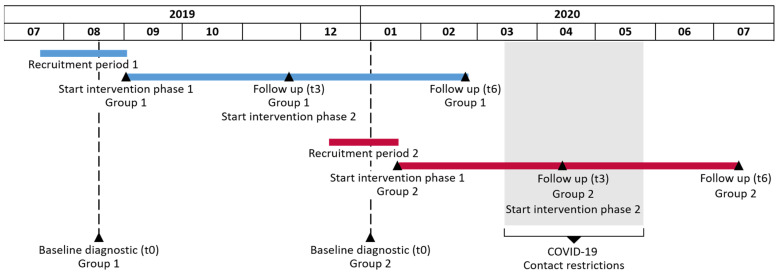
Study timeline. Blue lines represent relevant time points of the first recruitment group, red lines of the second recruitment group. The grey box shows the time of the COVID-19 contact restrictions in Germany. During this time no on-site appointments were possible.

**Figure 2 ijerph-19-09469-f002:**
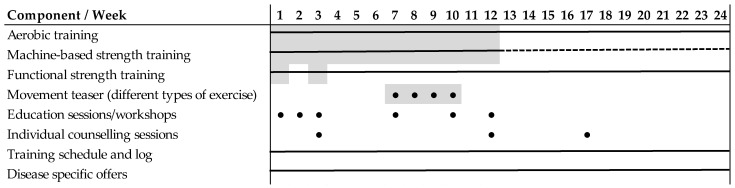
‘Multipill-Exercise’ intervention components. Greyed: supervised exercise sessions. Dotted lines: optional. Dots display the special intervention components respective to the week they took place.

**Figure 3 ijerph-19-09469-f003:**
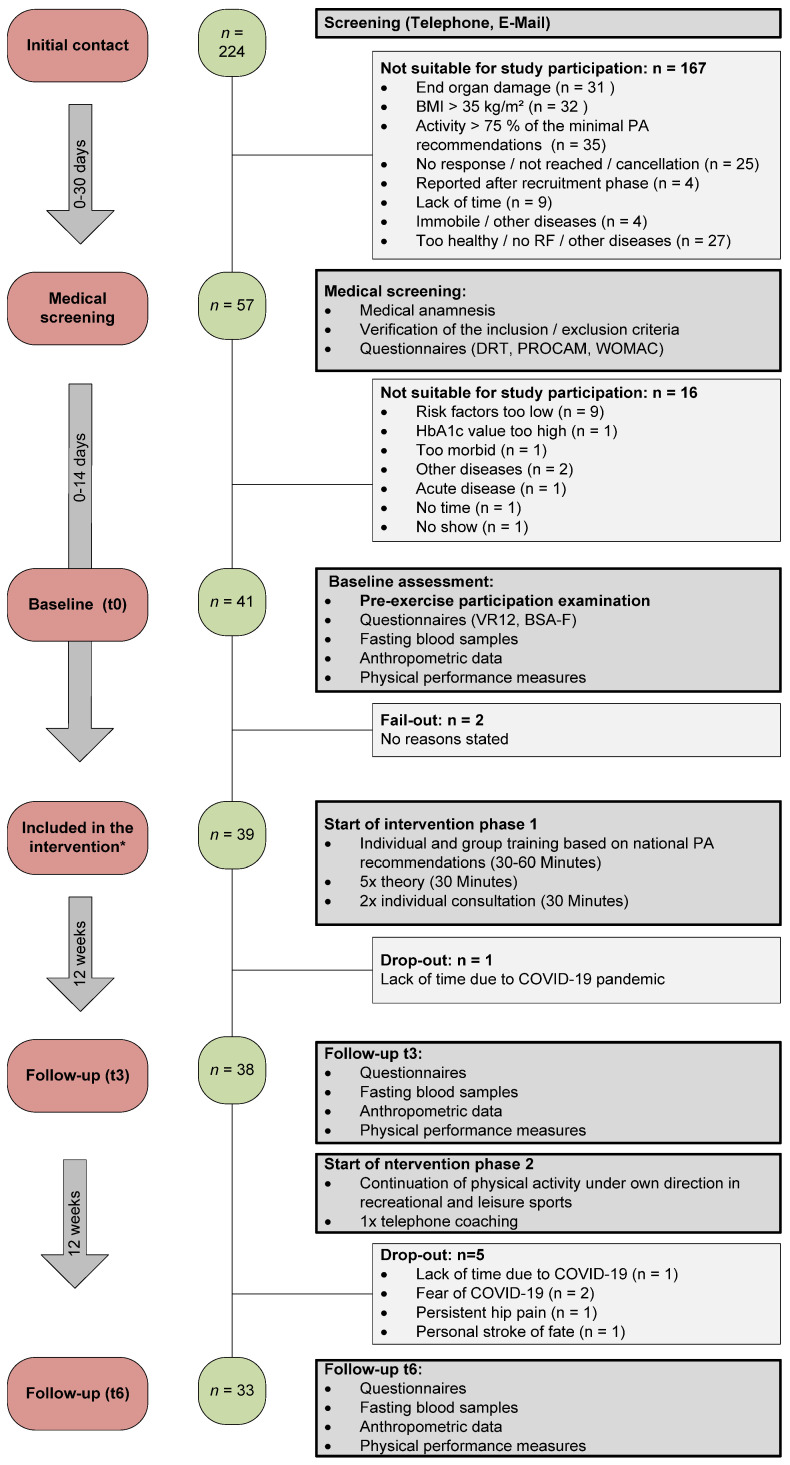
Study Flow Chart of the MultiPill-Exercise intervention summarizing the total sample from recruitment to the end of the intervention. PA: Physical activity; RF: Risk factor; DRT: Diabetes Risk Test; PROCAM: Prospective Cardiovascular Münster Study Score; WOMAC: Western Ontario and McMasters Universities Osteoarthritis Index; VR12: Veterans Rand 12; BSA-F: Physical activity, exercise and sport questionnaire. * Study inclusion was defined as attendance of the first appointment of the intervention.

**Figure 4 ijerph-19-09469-f004:**
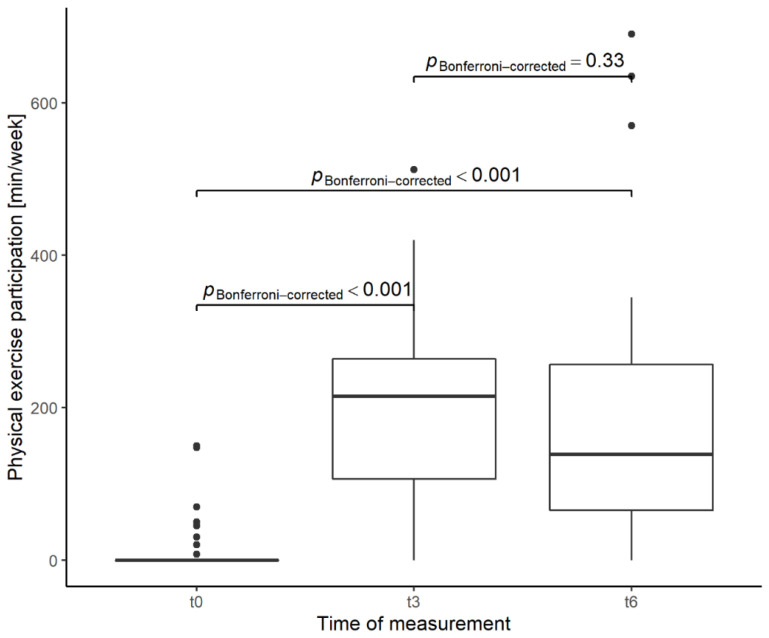
Post-hoc analysis of the total sample (*n* = 39) for the time effects t0 vs. t3 and t0 vs. t6 as well as t3 vs. t6 of physical exercise participation.

**Table 1 ijerph-19-09469-t001:** Eligibility criteria for study participation displaying the inclusion and exclusion criteria.

Inclusion Criteria (Two or More of the Following Diagnoses and/or Risk Profiles)		
	Diagnosis	Risk profile
Osteoarthrosis of hip and/or knee	According to the ACR criteria ^i^	WOMAC > 15 (pain and function)
Diabetes mellitus Type 2	HbA1c < 6.5% Fasting blood glucose < 126 mg/dL	German diabetes risk score ≥ 57 points
Cardiovascular disease	Including hypertension, coronary artery disease, arteriosclerosis, etc.	PROCAM-Score > 1.2 at risk (compared to gender and age group)
Overweight/Obesity	BMI ≥ 27–≤ 35 kg/m^2^	
**Exclusion criteria**		
Overall	- Physical activity > 75% of the (inter-) national physical activity recommendations ^ii^ - End-organ damage ^iii^ - Clinically significant deviations in certain laboratory parameters (including liver values above two times of the norm (AST, ALT, y-GT) as well as blood glucose and-HbA1c values beyond the inclusion criteria) - Medications that constitute an exclusion criterion according to the study physician - Clinical findings in the medical screening or pre-participation examination for physical exercise capacity, that precluded participation in the study	
Osteoarthrosis	Appointment for elective joint replacement	

^i^ Differentiation by arthrosis type (include all): post-traumatic, inflammatory, genetic/idiopathic, ^ii^ 150 min of moderate or 75 min of vigorous physical activities per week or an equivalent combination of both and twice a week muscle strengthening exercises, ^iii^ Osteoarthrosis: Artificial joint replacement; Diabetes mellitus Type 2: Kidney degradation requiring dialysis, severe retinopathy; CVD: Myocardial infarct, Stent, bypass; Overweight: Gastric band, AST: aspartate transferase; ALT: alanine transferase, y-GT: y-glutamyltransferase; ACR: American College of Rheumatology; BMI: Body mass index; HbA1c: glycated hemoglobin; WOMAC: Western Ontario and McMaster Universities Arthritis Index.

**Table 2 ijerph-19-09469-t002:** Participants’ characteristics at baseline including demographics and prevalence of chronic diseases including the total sample (*n* = 39).

Baseline Data	
*N* (thereof female)	39 (27)
Age (years)	
Mean ± SD	55.2 ± 10.3
Minimum (years)	27
Maximum (years)	69
Employed	30 (77)
Retired	8 (20)
In education	1 (3)
Physician	4 (10)
Medical assistant	9 (23)
Craftsperson	1 (3)
Freelancer	3 (8)
Civil cervices employee	9 (23)
Office employee	10 (26)
Others	3 (8)
BMI (Mean ± SD)	31.1 ± 3.0
Overweight/Obesity, *n* (%)	38 (97)
thereof > 27 kg/m^2^ < 30 kg/m^2^	11
thereof ≥ 30 kg/m^2^	24
Cardiovascular disease (yes, *n* (%)/risk factor, *n* (%))	28 (72)/2 (5)
thereof arterial hypertension	28/2
thereof pharmacological treatment	23
Diabetes mellitus type 2 (yes, *n* (%)/risk factor, *n* (%))	4 (10)/27 (69)
Osteoarthritis (yes, *n* (%)/risk factor, *n* (%))	17 (43)/11 (28)
thereof hip osteoarthritis	2/3
thereof knee osteoarthritis	15/8
Physical exercise participation (minutes/week)	
*Mean ± SD*	27.7 ± 94.3
*Median ± IQR*	0.0 ± 0.0
Minimum/Maximum	0.0/559.0 *

* The maximum value is due to a previous holiday and the participant stated that this did not represent a normal week.

**Table 3 ijerph-19-09469-t003:** Time comparisons for physical exercise participation from the mixed model estimates with imputed data.

Study Visit	Rank Means ^i^	RTE ^ii^	*p*-Value
Baseline	29.3	0.25	
t3 (12-weeks post baseline)	77.1	0.66	<0.001
t6 (24-weeks post baseline)	70.6	0.60	<0.001

ⁱ Rank Means were calculated and comparisons between time points were made from the mixed model. ⁱⁱ RTE: Relative treatment effect = a randomly chosen observation from the whole dataset results in a smaller value than a randomly chosen observation from the measurement time point with an estimated probability (in %) of RTE × 100.

**Table 4 ijerph-19-09469-t004:** The effectiveness of ‘MultiPill-Exercise’ intervention on physical performance measures, metabolic outcomes and health-related quality of life based on mixed model analysis.

Study Visit	Mean (SE)				Mean Difference (95% CI)		*p*-Value	ES ^i^
	Norm Values ^ii^	Baseline	t3	t6	Time Points			
**Physical Performance Measures**								
**Cardio-respiratory Outcomes**								
Relative VO_2_peak (mL/kg/min)	♀: 26.2 ♂: 32.8	20.0 (0.8)	22.7 (0.9)	22.0 (0.8)	t3-t0	2.6 (1.4 to 3.6)	<0.001	0.53
					t6-t0	2.0 (0.7 to 3.2)	0.001	0.41
VO_2_peak (l/min)	n.a.	1.8 (0.1)	2.0 (0.1)	2.0 (0.1)	t3-t0	0.2 (0.1 to 0.3)	0.001	0.32
					t6-t0	0.1 (0.01 to 0.3)	0.03	0.16
**Muscular Strength Measures**								
Maximum force: Extension (Nm)	n.a.	114.5 (8.7)	126.2 (9.1)	132.6 (8.9)	t3-t0	11.7 (2.7 to 20.7)	0.007	0.22
					t6-t0	18.1 (7.7 to 28.4)	<0.001	0.33
Maximum force: Flexion (Nm)	n.a.	88.4 (6.5)	91.5 (6.7)	96.1 (6.6)	t3-t0	3.2 (−4.4 to 10.7)	0.90	
					t6-t0	7.8 (−1.1 to 16.6)	0.10	
**Metabolic Measures (blood variables and anthropometrics)**								
Total Cholesterol (mg/dL)	130–190	217.6 (6.7)	215.0 (7.7)	210.5 (7.1)	t3-t0	−2.7 (−17.7 to 12.3)	1.00	
					t6-t0	−7.1 (−22.4 to 8.2)	0.75	
HDL-cholesterol (mg/dL)	≥35	58.7 (2.1)	59.9 (2.3)	61.5 (2.2)	t3-t0	1.1 (−2.3 to 4.6)	1.00	
					t6-t0	2.8 (−1.1 to 6.7)	0.23	
LDL-cholesterol (mg/dL)	≤160	148.2 (6.8)	146.6 (7.5)	141.4 (7.1)	t3-t0	−1.6 (−14.6 to 11.5)	1.00	
					t6-t0	−6.8 (−20.5 to 7.0)	0.68	
Triglycerides (mg/dL)	≤200	125.8 (10.8)	124.1 (11.7)	120.5 (11.1)	t3-t0	−1.7 (−19.8 to 16.4)	1.00	
					t6-t0	−5.2 (−23.0 to 12.5)	1.00	
Fasting glucose (mg/dL)	70–99	92.3 (1.8)	91.3 (2.2)	91.8 (1.9)	t3-t0	−1.0 (−6.4 to 4.4)	1.00	
					t6-t0	−0.5 (−4.8 to 3.9)	1.00	
HbA1c (%)	4.5–6.2	5.7 (0.1)	5.6 (0.1)	5.7 (0.1)	t3-t0	−0.1 (−0.2 to 0.1)	0.80	
					t6-t0	0.04 (−0.02 to 0.1)	0.32	
Body weight (kg)	n.a.	90.8 (2.0)	89.2 (2.0)	89.3 (2.0)	t3-t0	−1.6 (−3.3 to 0.02)	0.05	
					t6-t0	−1.6 (−3.6 to 0.5)	0.18	
**Health-related Quality of Life ^iii^**								
Physical Component Scale	40.7	42.7 (1.9)	44.3 (1.9)	44.2 (1.9)	t3-t0	1.6 (−1.3 to 4.4)	0.54	
					t6-t0	1.5 (−4.8 to 1.8)	0.80	
Mental Component Scale	53.1	47.0 (1.3)	50.1 (1.3)	46.3 (1.4)	t3-t0	3.1 (−0.4 to 6.6)	0.10	
					t6-t0	−0.7 (−4.6 to 3.2)	1.00	
General Health	64.4	44.6 (3.4)	60.8 (3.5)	58.0 (3.7)	t3-t0	16.2 (4.7 to 27.7)	0.003	0.76
					t6-t0	13.4 (3.0 to 23.8)	0.008	0.63

t3: (12-weeks post baseline); t6: (24-weeks post baseline), the analyses refer to *n* = 39, SE: standard error. CI: confidence interval; n.a.: not applicable, Means were calculated and comparisons between each measurement time points were made from the linear mixed model estimates. ^i^ ES: effect size (=intervention mean change/standard deviation at baseline), were only calculated for significant results. ^ii^ Norm values: VO_2_ according to Finger, et al. [47] referring to German citizens aged 55–64 years; Muscular strength values depend on age, sex and body weight; Metabolic measures according to the central laboratory of the University Hospital Tuebingen; Physical and Mental Component Scale according to Selim, et al. [48] referring to general US citizens mean age of 45 years, GH based on German citizens mean age 50 years according to Morfeld, et al. [49]. ^iii^ Health-related quality of life: Subscales Veterans Rand (VR) 12. Higher scores indicate better health-related quality of life (scale range 0–100) [48].

## Data Availability

The data presented in this study are available on request from the corresponding author.

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
