# Peer review of "Proof of Concept of a 6-Month Person-Oriented Exercise Intervention ‘MultiPill-Exercise’ among Patients at Risk of or with Multiple Chronic Diseases: Results of a One-Group Pilot Trial"

_ijerph, 2022, doi:10.3390/ijerph19159469_

Round 1

Reviewer 1 Report

I am satisfied with the improvement that the authors performed in their manuscript and I now consider that the manuscript may be accepted for publication in its current form.

Author Response

Dear Reviewer,

thank you for reviewing this manuscript. We very much appreciate that you consider the manuscript worthy of publication. Due to divergent reviewer comments, we have again made extensive changes, especially in the description of the intervention (chapter 2.4). We hope that the revised manuscript still meets your requirements. 

Best regards. 

Simone Schweda 

Reviewer 2 Report

While some corrections and improvements have been made to the paper, the basic flaws remained the same as in the original version. Unfortunately these are inherent characteristics of the study and cannot be corrected by making some changes in the text of the manuscript.

The volunteer bias and the very heterogeneous, "accidental" nature of the multimorbid group highly reduce the value of the study for the scientific community.

Author Response

Dear Reviewer, Thank you for reviewing this manuscript again. Based on the reviewer comments, we have again made extensive changes, especially in the description of the intervention (chapter 2.4) and the presentation of the results. We hope we were able to contribute to the improvement of the manuscript through the revisions. As outlined above, the study population is based on the common risk factor of systemic chronic inflammation, which is present in all included diseases. Subgroup analysis by disease combination or gender, age, social status were not possible due to the sample size. However, we would like to reiterate that the study is a pilot study that served as proof of concept for a larger study. In the follow-up study, we will be pleased to consider your comments regarding the study population. 

Kind regards, 

Simone Schweda 

Reviewer 3 Report

Dear Authors,

Dear Editors,

thank you for the occasion to review this interesting manuscript, I am happy to help improve further this paper with my comments.

This paper provides marked novelty and was set up with a diligently created design for the interventional period and testing; and can add value to the current literatur available; congratulations!

Nice figures; some could have better quality; some are confusing as sometimes seems other/not-matching info with explanation from text - has to be fixed/matched togeher

However, some weakness (mild to marked) generally in academic writing (guessing this is a PhD-candidate and novice to this), and also in Abstract, Method and Results has to be fixed by a major and very concenrated revision - I am convinced that the team of authors will succeed after focused revision elaborated! Go for it :-)

Most importantly, the main questions to me seems left without answer (see title, goal): miss 1 primary and main outcome: did the ones with more exercise do better/have better health status/outocomes vs. the low-performers in your multi-pill trial? - to me, being motivated to follow all this - the text flow and structure is not always conscise and often leads me (perceived) away from the main message/mission of this paper;

Abstract - academic writing! Mix of Method and Results - some info lacking

References - manuscript is based on well-chosen and a sound body of science

Intro - is really good and based on sound theory; only GAP is lacking to bridge from here to the AIM ("Therefore, the purpose .... is...") is much too long -> compact and more conscise; and Hypothesis in present version goes too far, too - again academic writing

Method - nice figures; overall diligently developed study and instrument, much work and basically congrats on this; however, some info is lacking or unclear or incomplete (or just "drowned" or was put into background to the massive overall flood of info) and has to be clarified, presented more detailed where necessary considering the MAIN GOAL; in addition, a total of 6-7 pages of method only is by far too much for such a paper! -> it seems a bit more a study protocol full methodology than a method section for a original paper; all the more as chapter 2.5 with 2.5 pages is much too long and then refers to 2 massive appendix tables - even a very intersted reader like me is loosing motivation to read further; info is lacking, eg. what kind(s) of lab-testing protocol did you follow for incremental testing, treadmill or cycle-ergometer? how you make sure that patients are able to translate PE program from the lab 110% or 90%  PO and/or 30 Watts to their field training alone - ?? what devices did they use to guarantee for that? Appendix tables 1-2 are massive, optimize formatting eg by smaller font-size and other edits. also, data clearnce and also modelling of data analysis must not be mixed within the section 2.7 Data Analysis; Generally I d recommend to display whole method in 1 scheme or chart and make a mix of necesary text, nice figures and informative large tables in order not to loose focus like on these 6-7 pages of method - not to forget the large appendix tables for method

Results - starts with a mix of Data Clearance (= clearly Method!) but shall rather start with the introduciton of profile/characteristics of main sample enrolled, nothing else; here some weakness/mix from method section is tracing up to results section, regrettably; prifle of sample (3.2) to me shall be better elaborated and displayed in more detail (see also Table 2) rather than appendix material; sometimes mix with discussion (eg. 3.3) that must be fixed; check if all figues and tables as presented add meaningful (sometimes more sometimes less, has to be maximized for all!) to answer your research questions and cope with the title and purpose of your study; to me, sometimes hard to follow although being very interested: therefore, i miss 1 primary and main outcome, as: did the ones with more exercise do better/have better health status/outocomes vs. the low-performers - to me, being motivated to follow all this - still hard to put together the pieces as the text flow and structure often leads me (perceived) away from the main message/mission of this paper; even an interested reader is already tired after results section

Discussion - generally good; however, starts with - again - a mix-up with conclusion incl. future perspective(s) and has to be fixed! Again, confusion due to different wordings for method/intervetion, eg. was not clear to me from method explained that each group did 2 12 weeks, I though it was only 1 6 month intervention with in-between testing at t3 and 1 final/post-testing at t6 - all this different wording is confusing, has to be fixed throughout the paper and where it really belongs to! Limitations has to be listed completely and not only occasionally.

Conclusion - is ok, but can be strenghtened

Author Response

Dear Reviewer 3,

Thank you for your comprehensive and valuable review of our manuscript. In the following you will find the requested changes and additions. The amendments to the text are embedded in the respective context (chapter). Based on the comments, the manuscript was extensively revised and relevant amendments were made. This relates to, among other things, the structure of the abstract as well as the main text. Figure and Table headings have been revised. All changes in the text are tracked. I hope that I was able to implement your constructive comments adequately. Next to this we would like to refere to the attached PDF we included as well, where the single comments have been answered. 

Yours sincerely,

Simone Schweda 

Dear Authors, Dear Editors,

 thank you for the occasion to review this interesting manuscript, I am happy to help improve further this paper with my comments.

This paper provides marked novelty and was set up with a diligently created design for the interventional period and testing; and can add value to the current literatur available; congratulations!

Nice figures; some could have better quality; some are confusing as sometimes seems other/not-matching info with explanation from text - has to be fixed/matched together

Thank you for this valuable comment. We have done a lot of editing and restructuring of the text. We hope we are more precise now and do match the information presented in the text and in the figures.

However, some weakness (mild to marked) generally in academic writing (guessing this is a PhD-candidate and novice to this), and also in Abstract, Method and Results has to be fixed by a major and very concenrated revision - I am convinced that the team of authors will succeed after focused revision elaborated! Go for it :-)

Thank you for your comment on the structure of the manuscript. The whole manuscript was written in accordance with the STORBE checklist for observational studies. We have reordered some points as displayed in more detail in the commented pdf, however we did not rearrange all requested paragraphs as the manuscript is in accordance with STROBE. We hope you can understand this.

Most importantly, the main questions to me seems left without answer (see title, goal): miss 1 primary and main outcome: did the ones with more exercise do better/have better health status/outocomes vs. the low-performers in your multi-pill trial? - to me, being motivated to follow all this - the text flow and structure is not always conscise and often leads me (perceived) away from the main message/mission of this paper;

Thanks for this comment. To highlight the key result of the manuscript the aim of the paper in the abstract in the introduction was emphasized. As this study is, as you also pointed-out, a one-group pilot trial we refrained from attributing the changes in health parameters to the intervention. As hopefully now becomes clearer, this pilot study aims at a proof of concept. In particular, we are interested in the increase in physical activity, which is the primary outcome. All other changes are also presented over time, but due to the lack of a control group we cannot conclude whether this is due to the intervention.

Abstract - academic writing! Mix of Method and Results - some info lacking

In accordance to your comment we have done editing regarding the abstract. However the abstract was written in accordance with the STROBE Statement for oberservation studies in conference abstracts (see STROBE Statement—Items to be included when reporting observational studies in a conference abstract).

References - manuscript is based on well-chosen and a sound body of science

Thank you for this comment.

Intro - is really good and based on sound theory; only GAP is lacking to bridge from here to the AIM ("Therefore, the purpose .... is...") is much too long -> compact and more conscise; and Hypothesis in present version goes too far, too - again academic writing

Thanks for this comment. We edited the introduction, especially regarding the PA recommendations to achieve health benefits as well as the aim and hypothesis. See therefore page 2, line 103-107:

“The aim of this study was a proof of concept of ‘MultiPill-Exercise’. The primary objective was to evaluate self-reported levels of physical exercise participation throughout the intervention. In addition, cardio-respiratory fitness, muscular strength, health-related quality of life (HRQoL) and blood levels were evaluated. The proof of concept was further related to training adherence and safety of the intervention.”

Method - nice figures; overall diligently developed study and instrument, much work and basically congrats on this; however, some info is lacking or unclear or incomplete (or just "drowned" or was put into background to the massive overall flood of info) and has to be clarified, presented more detailed where necessary considering the MAIN GOAL; in addition, a total of 6-7 pages of method only is by far too much for such a paper! -> it seems a bit more a study protocol full methodology than a method section for a original paper; all the more as chapter 2.5 with 2.5 pages is much too long and then refers to 2 massive appendix tables - even a very intersted reader like me is loosing motivation to read further; info is lacking, eg. what kind(s) of lab-testing protocol did you follow for incremental testing, treadmill or cycle-ergometer? how you make sure that patients are able to translate PE program from the lab 110% or 90%  PO and/or 30 Watts to their field training alone - ?? what devices did they use to guarantee for that? Appendix tables 1-2 are massive, optimize formatting eg by smaller font-size and other edits. also, data clearnce and also modelling of data analysis must not be mixed within the section 2.7 Data Analysis; Generally I d recommend to display whole method in 1 scheme or chart and make a mix of necesary text, nice figures and informative large tables in order not to loose focus like on these 6-7 pages of method - not to forget the large appendix tables for method

We do agree on this. Thanks for helping to become more precise. The method section was shortened according to your valuable comments. Especially paragraph 2.5, now 2.4 has been rewritten. We would like to refer to the paragraph here see page 4-5, line 142-188.

The lab-testing protocol is described in the outcome section, see 2.5.3. Physical Performance Measures, page 6, line 227-228: “The incremental test-protocol started with 25 W for women and 50 W for men and increased 25 W for both women and men each 3 minutes until exhaustion.”. The control of the training prescription has been added to the intervention description, page 4, line 154-155: “Training intensity is monitored by individual heart rate responses or perceived exertion (BORG CR10 0-10 point [32]).”

We do agree that the Appendix tables are extensive. However, especially the table A1, is necessary to explain the comprehensive intervention MultiPill-Exercise, as the intervention is not self-explanatory. We have deleted Table B1 however. Relevant information on Responder and Non-responder were added in the main text (see Results, page 11, line 369-373: Overall, 24 (61 %) participants exercised in line or beyond the PA recommendations at t3. At t6, 15 (38 %) participants reported a physical exercise participation in line with the PA recommendations. Seven participants (18 %) reported not to perform any physical exercise after 6-month, while the remaining participants stated an increase in physical exercise participation compared to baseline”)

We also did changes in the chapter of Data analysis, see chapter 2.6. We now ordered this chapter as 2.6.1 Data analysis, separated for primary and secondary outcome, as well as 2.6.2 Handling of missing data (see page 7-8, line 272- 328). The chapter again is in accordance with STROBE.

Results - starts with a mix of Data Clearance (= clearly Method!) but shall rather start with the introduciton of profile/characteristics of main sample enrolled, nothing else; here some weakness/mix from method section is tracing up to results section, regrettably; prifle of sample (3.2) to me shall be better elaborated and displayed in more detail (see also Table 2) rather than appendix material; sometimes mix with discussion (eg. 3.3) that must be fixed; check if all figues and tables as presented add meaningful (sometimes more sometimes less, has to be maximized for all!) to answer your research questions and cope with the title and purpose of your study; to me, sometimes hard to follow although being very interested: therefore, i miss 1 primary and main outcome, as: did the ones with more exercise do better/have better health status/outocomes vs. the low-performers - to me, being motivated to follow all this - still hard to put together the pieces as the text flow and structure often leads me (perceived) away from the main message/mission of this paper; even an interested reader is already tired after results section

Thanks for your comments on results. Again, I would like to refer to the STROBE checklist. The results are presented in line with the checklist, see STROBE Item 13 ((a) Report numbers of individuals at each stage of study—eg numbers potentially eligible,examined for eligibility, confirmed eligible, included in the study, completing follow-up, and analysed; (b) Give reasons for non-participation at each stage; (c) Consider use of a flow diagram).

All figure and table headings have been checked and edited. As recommended Table B1 was deleted.

Table 2 displaying the baseline characteristic was edited. However we decided to delete the drop-out column, as we do agree that this does not need further emphasize. We did not display the characteristics separately for the recruitment groups are sex for several reasons. First of all we do not expect the recruitment groups to be any different and we did not differ between the recruitment groups (besides Table B1 on adherence values) as the 3-level model including recruitment groups did not display significant results, indicating that no group differences are to be expected. As this study sample does not allow sub-group analyses, we did not differ for males and females either.

Thanks for your example on point 3.3. This sentence was added as an explanation. As the statistical method is not common we aimed to add this at this point. However, as it is also explained in the data analysis section we deleted this sentence at this point.

To address your point on the primary outcome: Our primary outcome was not a responder analysis but the question whether the participants increase PA after the intervention period. This primary outcome is mentioned in the abstract as well as at the very beginning of the results section. However, we agree that the description is very long and does not focus on the statistical analysis of the primary outcome at first site as chapter 3.3 starts with many descriptive stats. We changed this paragraph to make it more precise. We would furthermore like to point out, that this is a one-group pilot trial and therefore we can not account health effects on the intervention as a control group is missing to verify this.

Discussion - generally good; however, starts with - again - a mix-up with conclusion incl. future perspective(s) and has to be fixed! Again, confusion due to different wordings for method/intervetion, eg. was not clear to me from method explained that each group did 2 12 weeks, I though it was only 1 6 month intervention with in-between testing at t3 and 1 final/post-testing at t6 - all this different wording is confusing, has to be fixed throughout the paper and where it really belongs to! Limitations has to be listed completely and not only occasionally.

Thanks for your comment. We edited the discussion section as well. For the beginning of the introduction I would like to refer to the STROBE Statement again. As this serves as a shot information and presenting of key results of the study. However, we did change several aspects as you suggested. For example, PA measurements was changed to the limitation section (s. page 19, line 525-534). We further hope that the changes in the method section help the understanding for the discussion chapter.

A new chapter was added for limitations. Relevant points regarding the lack of a control grou, the control of training intensity and the short follow-up period have been added amongst others.

Conclusion - is ok, but can be strengthened

Your helpful comments were considered in the conclusion. We deleted the section on the pre-exercise participation screening and added an introduction sentence (s. page 19, line 549-550: “At a political level, calls for conservative treatment methods for people with multimorbidity are increasing. This study contributes to gain further understanding regarding the delivery of exercise interventions for individuals with NCDs.”)

Round 2

Reviewer 2 Report

The authors made several corrections in the manuscript, but unfortunately the basic problems with the experimental design cannot be corrected at this phase.

Summary major concerns:

1. Novelty. As I demonstrated in my first review, there are papers in the literature which had dealt with physical activity programs in multimorbid patients. While those programs did not emphasize the multimorbidity aspect that strong as the current manuscript, they, however, undoubtedly involved multimorbid groups. Thus, this manuscript lacks the breakthrough related to innovation point of view.

2. Vounteer bias. Mentioning the possibility of volunteer bias under "Limitations" does not eliminate this bias. With the current experimental design the volunteer bias (and its effect) is not only possible but highly probable. This does not allow to draw valid conclusions. Because of the volunteer bias the results are essentially predictable.

3. Heterogeneous nature of the subjects. While the authors mention that presence of systemic chronic inflammation can be a common feature of the selected diseases, this is not enough to handle diseases with different pathomechanism, symptoms, etc. as a group. There are several other diseases with chronic inflammation, patienets with those diseases, however, were not involved in the study. The heterogeneity (together with the small ample size) prevents the reader to answer the question: to whom are the results applicable and valid?

In spite of the huge efforts the authors made to prepare the original manuscript and to make several corrections, these inherent problems could not be solved, and cannot be corrected without going back to the roots (study design). 

Author Response

Dear Reviewer, 

we regret that we could not convince you of the quality of this manuscript. 
However, we would like to thank you for your helpful remarks and comments on the manuscript. 

Yours sincerely 
Simone Schweda 

Reviewer 3 Report

Dear Authors,

your paper has much improved - well done! some minor edits I d recommend to elaborate before ready for consideration of publication, see my minor comments.

However, STROBE or even CONSORT criteria are not a duty protocol to follow but a guidelines, and as you correctly mentioned, is more prevalent as a guideline for conference documents rather that an original article; therefore, I understood the 1st time you menioned this, but is too easy and cheap to hide behind that, and even keeps you away from a massive improvement of any manuscript when you keep hiding behind such guidelines rather than improving your academic writing; however, this is a kind tip that you can take or leave.

Best and good success

Author Response

Reviewer 2:

Comments and Suggestions for Authors

Dear Authors,

your paper has much improved - well done! some minor edits I d recommend to elaborate before ready for consideration of publication, see my minor comments.

However, STROBE or even CONSORT criteria are not a duty protocol to follow but a guidelines, and as you correctly mentioned, is more prevalent as a guideline for conference documents rather that an original article; therefore, I understood the 1st time you menioned this, but is too easy and cheap to hide behind that, and even keeps you away from a massive improvement of any manuscript when you keep hiding behind such guidelines rather than improving your academic writing; however, this is a kind tip that you can take or leave.

Best and good success

Submission Date

30 May 2022

Date of this review

12 Jul 2022 09:54:10

Dear Reviewer,

thank you for the time you have again devoted to this work and for the helpful comments that have significantly improved this manuscript.

We have also gladly incorporated the additional minor comments. You will find our detailed reply in the following.

Comment 1, page 3 line 116: I cannot see (maybe my mistake) the outlines LACK and GAP still - what is MISSING and DOES NOT EXIST - is needed to mention for seeling your pilot trial!

We changed this section to (see page 3 line 116 ff):

To summarize, physical inactivity is a major risk factor for many NCDs and people suffering from multimorbidity are specifically vulnerable as they are even less active than the norm. As health care programs including physical exercise interventions primarily focus on single disease management, a holistic approach specifically designed for people being at risk or suffering from multimorbidity is needed. Therefore, the aim of this study was to conduct and to proof the concept of a person-oriented exercise intervention (‘Mul-tiPill-Exercise’) aiming to enable participants with multimorbidity to engage in regular exercise according to the PA guidelines.

Comment 2, page 11, line 465: good explanation in your coverletter-PDF-file, but to strenghten you need the refernce mentioned here, too

We added the referred reference [47): Lee, Y.-S.; Laffrey, S.C. Predictors of Physical Activity in Older Adults With Borderline Hypertension. Nursing Research 2006, 55, 110-120.

Comment 3, page 11, line 469: ok, so why mentioning with a sentence, remove all what is NOT in this paper; not clear why to mention (maye I misunderstand)

We deleted the sentence.

Comment 4, page 13, line 512: this is only descriptive or do you have additional stats to include as column?

We did not change anything here, as this is only descriptive. No further column was added therefore.

Comment 5, page 22, line 763: Academic writin: Limiations must not be and extra point in main structure as it is part (final part) of the discussion; i d suggest to name it if you indeed want to add a number like 4.1 or so; but also fine withouth any order-number

The numbering was changed to 4.1 as suggested as one possible option.

In future papers, we will try to further improve the writing style and not only consider the guidelines as a reference for the structure of the manuscripts.

Thank you for reviewing this manuscript.

Best regards,

Simone Schweda

This manuscript is a resubmission of an earlier submission. The following is a list of the peer review reports and author responses from that submission.

Round 1

Reviewer 1 Report

I found the manuscript very interesting, well written, and it seems to me that this could be the first chapter of a larger work; at least the relevance of this paper for itself seems to underestimate the overall picture of its impact on further outcomes.

The study design is adequate and ethical, and the schemes presented in Figures 1 and 2 have simplified its understanding for the readers, and the comprehension of the decrease in the sample size since recruitment. However, the following doubt arises: authors have collected data from the second group (recruitment period 2) but I am not sure if that that was used or not. Please clarify if the 39 cases include both groups or just group 1. On line number 252 authors state that “In addition, on-site assessed objective … had to be canceled for group 2” but then in Table B2 adherence values are reported for group 2 (at T3?). It would also be interesting to compare those groups for primary and secondary outcomes.

In table 2, the values of the mean and the median of physical exercise participation may be misleading for some readers. Perhaps authors could indicate the number, which is necessarily inferior to 10 (and also min-max values) for participants presenting positive values in that outcome variable.  

For figure 3, and due to the number of cases in analysis, it would be interesting to present an overlay box and dot plot identifying where are positive values at baseline placed at t3 and t6. Alternatively, and in this line of reasoning, a scatterplot with baseline values on the horizontal axis and t3&t6 at the vertical axis (or two scatterplots, figure 3A and 3B) and also a diagonal line would benefit the reader's interpretation of the results, presenting differences and correlations at the same time. If both groups (from recruitment 1 and 2) were in analysis, then greyscale could be used to distinguish them.

My main concern is related to lines 120 and 463, where the authors state that “Details are depicted in the timeline in 错误!未找到引用源” and “Results for all secondary outcomes are displayed in 错误!未找到引用源”, respectively, which is very strange.

Reviewer 2 Report

Dear Authors, thank you for submitted manuscript. The issue of your work hits full the inactivity of millenium people. The article is well organised, English is fine, the statistical analysis is simple and well related. 

Reviewer 3 Report

The study focuses on a physical activity intervention program in patients with multiple chronic diseases. This is a definitely important issue, affecting huge number of patients, since in the majority of patients more than one chronic non communicable diseases can be diagnosed.

While the authors put huge effort in the design and implementation (several parts of the publication are of high quality - I do not go into these details, but I recognized and acknowledged them), the study still has severe weaknesses.

Basic epidemiological textbooks deal with the so called "volunteer bias", which, according to the description of participant recruitment, affects this study as well. Those who apply for participation, based on newspaper advertisements, etc., do not represent the population of multimorbid patients. They are the most motivated ones, most probably having a compliance much better than the average.

The other flaw in patient selection is the high proportion of physicians and medical assistants, based on similar reasons. Altogether, the characteristics of the participants make it very difficult or even impossible to draw conclusions for the total population of multimorbid patients.

While the authors made power calculations, the case numbers did not make an appropriate comparison between t3 and t6 possible with respect to several outcome parameters. Analyzing the long-term outcomes, however, this should have been important. 

While the authors emphasize how unique there study is, I do not completely agree with this. In the literature one can find studies involving patients with more than one chronic condition. Cardiac rehabilitation oriented studies, for example, sometimes deal with multimorbid patients (e.g.: Yu et al: Long-term changes in exercise capacity, quality of life, body anthropometry, and lipid profiles after a cardiac rehabilitation program in obese patients with coronary heart disease. Am J Cardiol, 91 (2003), pp. 321-325). Studies focusing on diabetes with a reported average BMI values around 30 also indicate a multimorbid condition. Metabolic syndrome is also such a situation, and there are studies concerning metabolic syndrome and exercise. Naturally, the mentioned studies are not exactly the same as the current study (and often do not emphasize the multimorbidity component), but I definitely would not say that "this is one of the first physical exercise intervention studies on participants with multiple NCDs" (obviously this depends also on how "one of the first" is defined).

Besides the major concerns related to the study design, there are some less severe issues.

In the section "Materials and methods" the majority of references related to the detailed description of the "Multipill-Exercise" are German publications (while the authors try to give as much details on the program as possible, but for details they refer to German language publications).

The "certain laboratory values" under the exclusion criteria is not exact (in spite of giving a few examples in Table 1). In Table 1 the authors marked the "End organ damage" with superscript iii, but this was not resolved later.

Referring to figures in my pdf-file erroneously contained Chinese characters (Details are depicted in the timeline in 错误!未找到引用源。and Results for all secondary outcomes are displayed in 错误!未找到引用源), this might be a conversion error...

Beside body weight as anthropometric outcome parameter, it would have been useful to measure and describe body composition (fat content, muscle mass, lean body weight, etc.), since the potential muscle mass increase during the program might have made a possible body fat decrease unnoticed.

The authors decided not to measure the adherence for the second phase, because the sessions here were not supervised. It still would have been useful to see the self-reported adherence.

I do not see how osteoarthritis can be handled together with cardiovascular diseases and diabetes (etc.) since the symptoms and the caused limitations are so different (in spite of the existing connection between musculosceletal diseases and obesity). It does not seem to be a good strategy to include such an "outlier" disease, since it makes the interpretation of results more difficult and uncertain.

Besides diagnosed diseases allowing high-risk conditions, makes the patient group even more "undefined" and accidental.